# Low Bone Mineral Density and Calcium Levels as Risks for Mortality in Patients with Self-Discontinuation of Anti-Osteoporosis Medication

**DOI:** 10.3390/ijerph19010197

**Published:** 2021-12-24

**Authors:** Chun-Sheng Hsu, Shin-Tsu Chang, Yuan-Yang Cheng, Hsu-Tung Lee, Chih-Hui Chen, Ya-Lian Deng, Chiann-Yi Hsu, Yi-Ming Chen

**Affiliations:** 1Department of Physical Medicine and Rehabilitation, Taichung Veterans General Hospital, Taichung 40705, Taiwan; chincent@gmail.com (C.-S.H.); ccdivlaser1959@gmail.com (S.-T.C.); yycheng@vghtc.gov.tw (Y.-Y.C.); 2College of Medicine, National Chung Hsing University, Taichung 402202, Taiwan; sdlee@vghtv.gov.tw; 3Department of Physical Medicine and Rehabilitation, Tri-Service General Hospital, School of Medicine, National Defense Medical Center, Taipei 11490, Taiwan; 4Department of Physical Medicine and Rehabilitation, Kaohsiung Veterans General Hospital, Kaohsiung 813414, Taiwan; 5School of Medicine, College of Medicine, National Yang Ming Chiao Tung University, Taipei 11221, Taiwan; orth.chen@gmail.com; 6Graduate Institute of Medical Sciences, National Defense Medical Center, Taipei 11490, Taiwan; 7Cancer Prevention and Control Center, Taichung Veterans General Hospital, Taichung 40705, Taiwan; 8Department of Neurosurgery, Neurological Institute, Taichung Veterans General Hospital, Taichung 40705, Taiwan; 9Department of Orthopedics, Taichung Veterans General Hospital, Taichung 40705, Taiwan; 10Department of Nursing, Taichung Veterans General Hospital, Taichung 40705, Taiwan; cfalsm@vghtc.gov.tw; 11Department of Medical Research, Taichung Veterans General Hospital, Taichung 40705, Taiwan; chiann@vghtc.gov.tw; 12Division of Allergy, Immunology and Rheumatology, Taichung Veterans General Hospital, Taichung 40705, Taiwan; 13Rong Hsing Research Center for Translational Medicine, National Chung Hsing University, Taichung 402202, Taiwan; 14Ph.D. Program in Translational Medicine, National Chung Hsing University, Taichung 402202, Taiwan

**Keywords:** osteoporosis, bone mineral density, mortality, serum calcium, adherence

## Abstract

Bone mass density (BMD) has been used universally in osteoporosis diagnosis and management. Adherence to anti-osteoporosis medication is related to mortality risk. This study aimed to investigate the relationship between mortality and low BMD of the femoral neck and vertebra among patients self-discontinuing anti-osteoporosis medication. Between June 2016 and June 2018, this single-center retrospective study recruited 596 participants who self-discontinued anti-osteoporosis medication. Patients were categorized into four groups by BMD of the right femoral neck and lumbar spine. Occurrence and causes of mortality were obtained from medical records. Independent risk factors and the five-year survival of various levels of BMD were analyzed by Cox regression and the Kaplan–Meier survival analysis. BMD value and serum calcium level were significantly lower in the mortality group (*p* < 0.001). Compared to the reference, the adjusted hazard ratio (HR) for all-cause mortality in patients with lower BMD of both the lumbar spine and femoral neck was 3.03. The five-year cumulative survival rate was also significantly lower (25.2%, *p* < 0.001). A low calcium level was also associated with mortality (HR: 0.87, 95% CI: 0.76–0.99, *p* = 0.033). In conclusion, lower BMD and calcium levels were associated with higher mortality risk in patients with poor adherence. Hence, patients self-discontinuing anti-osteoporosis medication should be managed accordingly.

## 1. Introduction

Osteoporosis is a systemic skeletal disorder characterized by decreased bone mineral density (BMD) and the microarchitectural deterioration of bone tissue, resulting in increased fracture risk. Osteoporotic fractures are widely suggested to be the most serious result of osteoporosis and are associated with increased morbidity and mortality, which incur a heavy financial cost worldwide [1,2]. Hence, the prevention and treatment of osteoporosis in elders is an important issue.

The risk of mortality after osteoporotic fractures is related to several factors, such as age, gender, type of fracture, fracture number, BMD, comorbidities, lifestyle, and anti-osteoporosis medication [2,3]. Increasing age is the main factor associated with increased mortality, regardless of the fracture site. For example, the risk of death is two-fold to four-fold higher in patients aged 80 years or older after hip fracture [2]. Regardless of the types of fracture (vertebral fracture or hip fracture),a higher mortality risk is noticeably in both women and men, especially after hip fracture [3,4]. Hip fractures are the most severe osteoporotic fractures [5] and, furthermore, lead to excess mortality risk, which has been estimated to be at least twice as high as that in the general population in an age-matched case-control analysis [6]. Compared to women, mortality is higher in men after hip or vertebral fracture [3,7]. Additionally, comorbidities (such as hypertension, diabetes mellitus, or chronic obstructive pulmonary disease) and previous health status have been associated with increased mortality in patients with either a vertebral fracture or hip fracture [3]. Poor adherence to anti-osteoporosis medication is also important as it has been linked with higher mortality risk following osteoporotic fractures [8]. A substantial proportion of patients may discontinue anti-osteoporosis medication [9,10], including self-discontinuing and bisphosphonate drug holidays. Mortality risks among patients continuing to take bisphosphonate are similar to those of patients on bisphosphonate drug holidays [11]. However, the factors contributing to mortality risks in patients who self-discontinued anti-osteoporosis medication remain largely unknown.

Spine and femoral neck BMD are a standard diagnostic assessment in osteoporosis, which is usually measured using densitometry applications, such as dual X-ray photon absorptiometry [12]. Decreased BMD was confirmed as a major risk factor for osteoporotic fractures [2]. It can also be used to evaluate the therapeutic efficacy of anti-osteoporosis medication [13,14]. Nonetheless, prior studies mainly investigated the relationship between low BMD and osteoporotic fractures. Only two previous studies assessed the direct correlation between low BMD and mortality in elderly women and stroke patients [15,16]. To date, there have been no studies on the relationship between BMD and mortality risks in patients discontinuing anti-osteoporosis medication. Moreover, in the two aforementioned studies, BMD was measured at the distal radius, proximal radius, and calcaneus [15,16]. Hence, there are no data on the relationship between low BMD measured at the femoral neck and vertebra and mortality.

The aim of this study was to investigate the relationship between mortality and low BMD of the femoral neck and vertebra among patients who self-discontinued anti-osteoporosis medication in the Taiwanese population.

## 2. Materials and Methods

### 2.1. Enrolled Participants

This was a single-center, retrospective study conducted at Taichung Veterans General Hospital, Taiwan. Our osteoporotic database enrolled osteoporotic patients that identified through the presence of the International Classification of Disease, 9th Revision, Clinical Modification (ICD-9-CM) and 10th edition (ICD-10) diagnostic code for osteoporosis (733.0, 733.00, 733.01, 733.02, 733.03, 733.09/M80–82) or for osteoporotic fractures, including vertebral fractures (805.2–805.9/S22, S32), hip fractures (820.x/S72), humeral (812.x/S42), and radio-ulnar fractures (813.x/S52, S62) on medical records. Patients with a history of traumatic accidents were excluded. In this study, we enrolled 1192 participants, aged 45 years old or older from our osteoporotic database, with prior exposure to anti-osteoporosis medication that was discontinued between June 2016 and June 2018. Among them, 596 patients who were lost to follow-up, lacked BMD data, discontinued by physician recommendation as drug holidays for bisphosphonates, or who took anti-osteoporosis medication in other hospitals were excluded from this study. This study was approved by the Ethics Committee of Clinical Research, Taichung Veterans General Hospital (CF18067B). As patient data were anonymized before analysis, the requirement for written consent from patients was waived for this study and approved by the Ethics Committee.

### 2.2. Study Design

Patients with measured BMD and prior exposure to anti-osteoporosis medication that was discontinued between June 2016 and June 2018 were enrolled in this study. In this study, denosumab discontinuation was defined as the absence of denosumab claim more than 6 months + 8 weeks since the last dose and without any anti-osteoporosis medication other than denosumab within 6 months + 8 weeks after prior denosumab treatment. We defined bisphosphonates or raloxifene discontinuation as ≥12 months without any anti-osteoporosis medication prescription claims after prior bisphosphonate or raloxifene treatment. Teriparatide discontinuation was defined as the absence of any anti-osteoporosis medication more than 8 weeks after a prior dose of teriparatide, and the duration of teriparatide after the first dose was less than 18 months. The occurrence of mortality and causes of death were obtained via telephone interview with the patients’ family and from health records maintained by the Taiwan Ministry of Health and Welfare (TMHW). Demographic data, clinical fracture risks (such as alcohol usage and smoking history), sites of fractures, number of fractures, BMD, prior anti-osteoporosis medication before discontinuation, comorbidities, and laboratory tests were then traced retrospectively from the electronic health records. The normal range of serum calcium, phosphorus, and intact-PTH were defined as 8.4–10.2 mg/dL, 2.5–4.5 mg/dL, and 15–68.3 pg/mL, respectively. Additionally, patients who received anti-osteoporotic treatment were identified. The anti-osteoporotic medications were categorized according to the last prescription of bisphosphonates (alendronate, ibandronate, and zoledronic acid), selective estrogen receptor modulator (raloxifene), recombinant human parathyroid hormone (teriparatide), and receptor activator of nuclear factor k-B ligand (RANKL) inhibitor (denosumab).

### 2.3. Fracture Identification

The data on osteoporotic fracture occurrence and fracture sites were retrospectively extracted from Taichung Veterans General Hospital’s medical records and radiographic reports. Osteoporotic fractures included vertebral fractures, hip fractures, distal radius fractures, such as Colles’ fractures, and fractures of other bones, such as the proximal humerus, ribs, tibia-fibula, patella, and pelvis. Participants with fractures due to vehicular accidents or high-impact trauma (ICD-9-CM code E810–E819, E881–E883, E884), pathological fractures (733.14, 733.15/M84), and those with a diagnosis of Paget’s disease (731.0/M88) were excluded for further analysis. The number of fractures was calculated by a summation of the above-mentioned fracture events.

### 2.4. BMD Measurements

BMD measurements of the lumbar spine (L1–L4) and bilateral femoral necks were obtained through dual-energy X-ray absorptiometry (DXA), using the Lunar Prodigy (General Electric, Fairfield, CT, USA), with the results expressed in g/cm^2^. The least significant change was ±0.010 g/cm^2^ for the lumbar spine (L1–L4) and ±0.012 g/cm^2^ for the femoral neck. T-scores were determined according to the manufacturer’s reference data. BMD was measured before using anti-osteoporotic medications.

Spine (≥0.715 g/cm^2^ vs. <0.715 g/cm^2^) and right femoral neck BMD (≥0.550 g/cm^2^ vs. <0.550 g/cm^2^) were classified according to the lower tercile and higher two-thirds distribution. Group A comprised patients with BMD of lumbar spine ≥ 0.715 g/cm^2^ and right femoral neck ≥ 0.550 g/cm^2^. Group B consisted of patients with BMD of lumbar spine < 0.715 g/cm^2^ and right femoral neck ≥ 0.550 g/cm^2^. Group C patients had BMD of lumbar spine ≥ 0.715 g/cm^2^ and right femoral neck < 0.550 g/cm^2^. Group D had BMD of lumbar spine < 0.715 g/cm^2^ and right femoral neck < 0.550 g/cm^2^.

### 2.5. Statistical Analysis

The demographic data of the continuous parameters are shown as mean ± standard deviation and for the categorical variables as the number of patients. The Chi-Square test and the Kruskal–Wallis test were used to compare variables among patients by BMD of lumbar spine and right femoral neck. Comparisons of parameters between survivors and non-survivors were calculated by the Mann–Whitney U test and Fisher’s exact test. Cox regression analysis was used to investigate independent risk factors associated with mortality with the adjustment of age, gender, number of fractures, BMD, laboratory tests, and comorbidities. Kaplan–Meier survival analysis by gender was used to determine the 5-year patient survival of the various levels of BMD. All data were analyzed using the Statistical Package for the Social Sciences (SPSS) version 22.0. Significance was set at *p* < 0.05.

## 3. Results

### 3.1. Demographic Data and Concomitant Medication by BMD

Overall, 596 participants (mean follow-up time: 5.0 years ± 0.2 years) were enrolled in this study, and the majority of them were female (Table 1). BMD was classified according to tercile distribution (291 in group A; 108 in group B; 107 in group C; and 90 in group D). The mean ages for groups A, B, C, and D were 73.9 years, 78.6 years, 81.5 years, and 81.7 years, respectively. Lower BMD of the lumbar spine and right femoral neck (group D) was correlated with lower body height (*p* = 0.003), lower body weight (*p* < 0.001), lower body mass index (BMI) (*p* < 0.001), lower serum calcium level (*p* < 0.021), lower serum 25(OH)D3 level, and higher prevalence of stroke (*p* = 0.001). Osteoporosis risk factors including smoking and alcohol intake did not differ among the four groups.

### 3.2. Comparisons between Survivors and Non-Survivors

As shown in Table 2, there were 386 survivors and 210 deaths among the participants who had self-discontinued anti-osteoporosis medication. The mortality group had a higher prevalence of more than two fracture sites (survival group, 14.5% vs. mortality group, 28.6%; *p* < 0.001), a higher mean age (survival group, 74.4 years vs. mortality group, 82.7 years; *p* < 0.001), lower body weight, lower BMI, and higher prevalence of smoking. According to classification by BMD, in the survival group, the proportions of groups A, B, C, and D were 58.0%, 17.9%, 13.5%, and 10.6%, respectively; in the mortality group, the proportions of groups A, B, C, and D were 31.9%, 18.6%, 26.2%, and 23.3%, respectively. Patients from group D were more prevalent in the mortality group than in the survival group (23.3 vs. 10.6%). However, there was no statistically significant difference between the T-score and mortality (data not shown). Additionally, the mortality group had a lower serum 25(OH)D3 level, lower serum calcium level, higher serum creatinine level, higher serum iPTH level, and higher prevalence of comorbidity, such as diabetes mellitus, hypertension, and stroke, compared with the survival group.

### 3.3. Predictive Factors for Mortality in Patients with Prior Exposure to Anti-Osteoporosis Medication That Was Later Discontinued

To identify independent factors associated with all-cause mortality in the enrolled participants, the Cox regression analysis was performed (Table 3). Risk factors significantly associated with all-cause mortality were male sex, older age, lower BMD, lower serum calcium level, higher serum creatinine level, and a history of comorbid diabetes mellitus. In terms of BMD, the highest mortality risk was noted in group D. Compared to the reference (group A), the adjusted hazard ratios (HR) for mortality were 1.68 (95% confidence interval, CI, 0.93–3.04, *p* = 0.083) in group B; 1.90 (95% CI 1.13–3.14, *p* = 0.016) in group C; and 3.09 (95% CI 1.84–5.21, *p* < 0.001) in group D (Table 3). As BMD in the spine and right femur decreased, the mortality risk soared (*p* for trend < 0.001). Additionally, lower BMD of the right femoral neck had a higher impact on mortality than that of the lumbar spine (1.90 vs. 1.68).

### 3.4. Survival Analysis of Participants by BMD and Gender

The cumulative five-year survival rates of groups A, B, C, and D were 58.86%, 50.36%, 32.99%, and 25.18%, respectively (Figure 1A). The cumulative five-year survival rates of group A, B, C, and D in females were 70.28%, 59.82%, 36.93%, and 26.6%, respectively (Figure 1B) and 31.06%, 15.22%, 21.39%, and 56.25% in males, respectively (Figure 1C).

### 3.5. Causes of Mortality

Among the causes of fatality, 60 (23.4%) died due to cerebral-cardiovascular or metabolic diseases; 57 (22.3%) died from cancers; 36 (14.1%) died from infectious diseases; 18 (7.0%) died due to accident; 10 (3.9%) died from degenerative diseases, and 57 (22.3%) died from others causes (Figure 2).

## 4. Discussion

Our results suggest that lower BMD at the hip or spine is associated with different five-year mortality risks in patients with self-discontinuation of anti-osteoporosis medication. Moreover, lower BMD over both the femoral neck and spine was associated with the highest mortality risk with a hazard ratio of 3.09, and the cumulative survival rate was 25.18%. Compared to lower BMD of the spine, lower BMD of the right femoral neck had more impact on mortality.

Previous studies demonstrated that the main causes of mortality in osteoporotic populations included cardiovascular disease, diabetes mellitus, respiratory disease, and cancers [2,17,18]. These causes of death appeared to be similar to those seen in the general population. In our study, the main causes of death in those who self-discontinued anti-osteoporosis medication were consistent with prior reports.

The causes of death in patients with osteoporosis is related to a number of factors, such as age, gender, type of fracture, fracture number, BMD, comorbidities, lifestyle, and anti-osteoporosis medication [2,3]. Old age, male gender, hip fracture, a higher number of fractures, and more comorbidities (such as hypertension, diabetes mellitus, or chronic obstructive pulmonary disease) have been associated with increased mortality. Additionally, adherence to anti-osteoporosis medication is an important factor which affects mortality risk. In Taiwan, Yu et al. reported that all-cause mortality rates were higher in patients without good adherence to anti-osteoporosis medication [8]. Compared to patients with good adherence, there was a 10–18% increase in the short- and long-term mortality risk among those with poor adherence. Hence, risk assessments in patients without good adherence are vitally important. However, there are no reports in the literature on the causes of death and mortality risk in patients with poor adherence. Our findings demonstrated that low BMD of the spine and femoral neck was associated with a higher mortality risk in patients who have discontinued anti-osteoporosis medication.

BMD strongly correlates with fracture risk, and the risk increases by one and a half to three-fold for every 1 SD decrease in BMD [12,14,19]. Low BMD is considered not only a major risk factor for fractures, but also an important risk factor for death [15,20]. Low BMD is also related to general health status and has been linked with many chronic diseases [2]. In women, low BMD has been associated with increased mortality independent of fracture [18]. However, only two previous studies have analyzed the direct correlation between low BMD and mortality [15,16]. The populations included in these two studies were elderly women and stroke patients. The studies found that low BMD increased mortality risk, although all deaths in these studies were unrelated to fractures. However, BMD in these two studies was measured at the distal radius, proximal radius, and calcaneus, rather than at the femoral neck or vertebra bone. Previous studies reported a non-significant increase in mortality following forearm, non-hip, and non-vertebral fractures [2,3,20]. Hence, assessments of BMD of the femoral neck and spine may have more clinical significance with respect to mortality. However, no studies have investigated the relationship between BMD of the hip/spine and mortality risks in patients without good adherence to anti-osteoporosis medication. Our results show that low BMD of the spine or femoral neck worsened the mortality risk in participants discontinuing anti-osteoporosis medication after a five-year follow-up. Moreover, this association was more significant in the female participants. A three-fold increased mortality risk was observed in those with low BMD of both the spine and femoral neck. The five-year cumulative survival rate was 25.18% in those with BMD of the lumbar spine < 0.715 g/cm^2^ and right femoral neck < 0.550 g/cm^2^. Furthermore, compared to low BMD of the spine, low BMD of the right femoral neck had a greater impact on mortality. Our results are consistent with a previous study that hip fractures had a higher mortality risk than vertebral fractures [3]. This report indicated that in managing osteoporosis patients, measurement of BMD at the spine and femoral neck is warranted to predict mortality risks in osteoporotic patients with poor compliance.

Our study showed that a lower serum calcium level, lower serum 25(OH)D3 level, and higher iPTH level are related to higher mortality risk. Rossini et al. note that the measurement of serum calcium should be one of the first-line investigations in patients with osteoporosis [12]. However, Okyaya et al. reported levels of serum calcium and ionized calcium were not associated with the development of postmenopausal osteoporosis [21]. Slinin et al. found an association between higher serum calcium levels and an increased risk of cardiovascular events in postmenopausal women with osteoporosis [22]. There are scanty data in the literature on the relationship between serum calcium level and mortality in osteoporotic populations. The present study is the first to demonstrate that serum calcium levels is a risk for mortality compared with normo-calcemic patients. We hypothesized that low serum calcium levels could be associated with malnutrition or critical illness [23]. Further studies are needed to confirm our findings.

To the best of our knowledge, this is the first study to investigate mortality risks among patients who have self-discontinued anti-osteoporosis medication. However, this study had some limitations. First, the number of male participants in our study was relatively small; therefore, it might be underpowered to detect survival differences among groups in the male population. However, we have provided valuable information to guide the health management strategy for patients discontinuing anti-osteoporosis medication. Second, this was a retrospective study and thus a causal relationship could not be drawn. Recall bias, selection bias, and under-detection of fractures may exist in this study. Third, the reasons for discontinuing anti-osteoporosis medication, mortality, patient nutritional factors, and the genetic predisposition were not provided in this study. Good adherence is significantly associated with survival benefits in patients with osteoporosis [8], and hence, further studies are needed to investigate the factors influencing adherence to anti-osteoporosis medication.

## 5. Conclusions

The results of this real-world, hospital-based study demonstrate that lower BMD of the spine/femoral neck and lower calcium level were associated with mortality risks in patients with self-discontinuation of anti-osteoporosis medication. Further studies are needed to determine whether correction of these factors might minimize mortality risks in osteoporosis patients.

## Figures and Tables

**Figure 1 ijerph-19-00197-f001:**
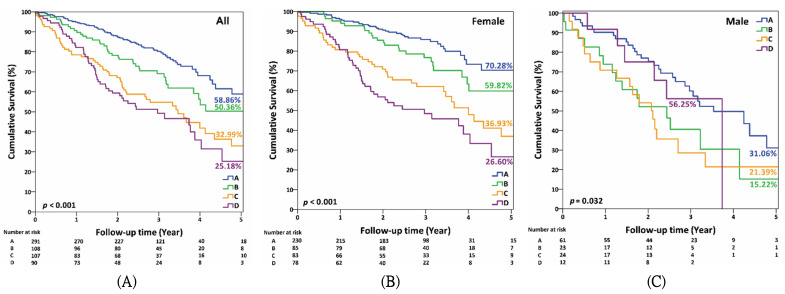
Kaplan–Meier curve of 5-year mortality rates by BMD of spine and right femoral neck in (**A**) all, (**B**) female, and (**C**) male participants. Group A: BMD of lumbar spine ≥ 0.715 g/cm^2^ and right femoral neck ≥ 0.550 g/cm^2^. Group B: BMD of lumbar spine < 0.715 g/cm^2^ and right femoral neck ≥ 0.550 g/cm^2^. Group C: BMD of lumbar spine ≥ 0.715 g/cm^2^ and right femoral neck < 0.550 g/cm^2^. Group D: BMD of lumbar spine < 0.715 g/cm^2^ and right femoral neck < 0.550 g/cm^2^. Comparisons of survival curves by log-rank test. Figure (**A**): all pairwise *p* < 0.05 except C vs. D; Figure (**B**): all pairwise *p* < 0.05 except A vs. B and C vs. D; Figure (**C**): A vs. B and A vs. C, *p* < 0.05.

**Figure 2 ijerph-19-00197-f002:**
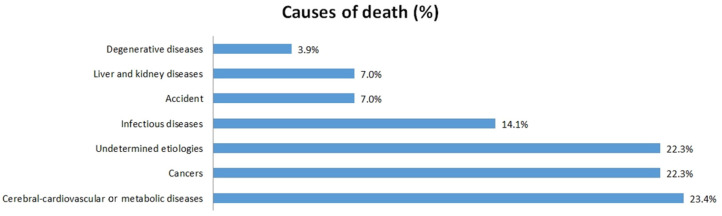
Causes of death in patients who had discontinued anti-osteoporosis medication.

**Table 1 ijerph-19-00197-t001:** Demographic data of participants by BMD classified according to tercile distribution.

Variables	Group A Lumbar Spine ≥ 0.715 g/cm^2^ and Right Femoral Neck ≥ 0.550 g/cm^2^	Group B Lumbar Spine < 0.715 g/cm^2^ and Right Femoral Neck ≥ 0.550 g/cm^2^	Group C Lumbar Spine ≥ 0.715 g/cm^2^ and Right Femoral Neck < 0.550 g/cm^2^	Group D Lumbar Spine < 0.715 g/cm^2^ and Right Femoral Neck < 0.550 g/cm^2^	*p* Value
	*n* = 291	*n* = 108	*n* = 107	*n* = 90	
Age (years)	73.9 ± 10.3	78.6 ± 9.0	81.5 ± 10.2	81.7 ± 8.4	<0.001 ^#,¤,à^
Gender-Female	230 (79.0%)	85 (78.7%)	83 (77.6%)	78 (86.7%)	0.368
Body height (cm)	155.2 ± 7.2	154.8 ± 7.4	154.7 ± 7.9	151.4 ± 8.9	0.003 ^à,ê,¢^
Body weight (kg)	57.7 ± 9.5	57.2 ± 11.7	54.3 ± 9.5	49.5 ± 10.9	<0.001 ^¤,à,ê,¢^
BMI (kg/m^2^)	24.0 ± 3.8	24.0 ± 4.4	22.7 ± 3.7	21.5 ± 4.1	<0.001 ^#,¤,ê^
Smoking	27 (9.4%)	11 (10.2%)	14 (13.3%)	9 (10.0%)	0.730
Alcohol consumption	16 (8.7%)	3 (4.0%)	5 (5.8%)	5 (7.4%)	0.539
Calcium supplement	180 (61.9%)	74 (68.5%)	79 (73.8%)	55 (61.1%)	0.106
Vitamin D supplement	55 (18.9%)	26 (24.1%)	29 (27.1%)	20 (22.2%)	0.316
Raloxifene	92 (31.6%)	32 (29.6%)	30 (28.0%)	34 (37.8%)	0.491
Alendronate	87 (29.9%)	25 (23.2%)	28 (26.2%)	26 (28.9%)	0.574
Zoledronic acid	23 (7.9%)	14 (13.0%)	12 (11.2%)	4 (4.4%)	0.140
Denosumab	123 (42.3%)	62 (57.4%)	58 (54.2%)	44 (48.9%)	0.025 ^#^
Teriparatide	31 (10.7%)	19 (17.6%)	24 (22.4%)	35 (38.9%)	<0.001 ^¤,à,ê,¢^
Laboratory test					
25(OH)D3 (ng/mL)	31.3 ± 13.4	25.0 ±6.1	15.1 ± 9.3	14.8 ± 10.2	0.036
Calcium (mg/dL)	8.8 ± 1.0	8.6 ±1.3	8.6 ± 1.1	8.4 ± 1.5	0.021¤
Phosphorus (mg/dL)	3.5 ± 1.2	3.4 ±1.0	3.4 ± 1.2	3.5 ± 1.3	0.932
Creatinine (mg/dL)	1.0 ±0.8	0.9 ± 0.3	1.3 ± 1.2	1.3 ± 1.7	0.179
Intact-PTH (pg/dL)	144.2 ±167.7	83.4 ± 81.6	276.5 ± 372.9	389.8 ± 591.1	0.108
Comorbidity					
Osteoarthritis	201 (69.1%)	80 (74.1%)	78 (72.9%)	50 (55.6%)	0.023 ^à,ê,¢^
Rheumatoid arthritis	19 (6.5%)	13 (12.0%)	10 (9.4%)	5 (5.6%)	0.232
Diabetes mellitus	76 (26.1%)	33 (30.6%)	45 (42.1%)	19 (21.1%)	0.005 ^¤,¢^
Hypertension	146 (50.2%)	62 (57.4%)	75 (70.1%)	59 (65.6%)	0.001 ^¤,à^
Stroke	41 (14.1%)	19 (17.6%)	29 (27.1%)	27 (30.0%)	0.001 ^¤^
Cataract	85 (29.2%)	31 (28.7%)	31 (29.0%)	23 (25.6%)	0.926
Hyperthyroidism	22 (7.6%)	10 (9.3%)	11 (10.3%)	3 (3.3%)	0.286
Chronic liver disease	61 (21.0%)	19 (17.6%)	28 (26.2%)	14 (15.6%)	0.255

Continuous data were expressed in mean ± SD; Categorical data were expressed in number and percentage. *p* value by Kruskal–Wallis test. Posthoc analysis by Dunn–Bonferroni test. *p* < 0.05: ^#^ A vs. B; ^¤^ A vs. C; ^à^ A vs. D; ^ê^ B vs. D; and ^¢^ C vs. D. BMI, body mass index.

**Table 2 ijerph-19-00197-t002:** Comparisons of demographic data of participants by survival status.

Variables	Survival (*n* = 386)	Mortality (*n* = 210)	*p* Value
Number of fractures			<0.001
0	65 (16.8%)	15 (7.1%)	
1	265 (68.7%)	135 (64.3%)	
≥2	56 (14.5%)	60 (28.6%)	
Age (years)	74.4 ± 10.1	82.7 ± 8.5	<0.001
Gender-Female	333 (86.3%)	143 (68.1%)	<0.001
Body height (cm)	154.5 ± 7.3	154.6 ± 8.4	0.844
Body weight (kg)	56.6 ± 10.3	54.4 ± 10.9	0.009
BMI (kg/m^2^)	23.8 ± 3.9	22.8 ± 4.2	0.005
Smoking	39 (7.6%)	32 (15.5%)	0.006
Alcohol consumption	15 (6.0%)	14 (8.5%)	0.428
Bone mineral density (g/cm^2^)			<0.001
A: Lumbar spine ≥ 0.715 and right femoral neck ≥ 0.550	224 (58.0%)	67 (31.9%)	
B: Lumbar spine < 0.715 and right femoral neck ≥ 0.550	69 (17.9%)	39 (18.6%)	
C: Lumbar spine ≥ 0.715 and right femoral neck < 0.550	52 (13.5%)	55 (26.2%)	
D: Lumbar spine < 0.715 and right femoral neck < 0.550	41 (10.6%)	49 (23.3%)	
Calcium supplement	242 (62.7%)	146 (69.5%)	0.114
Vitamin D supplement	77 (20.0%)	53 (25.2%)	0.165
Raloxifene	127 (32.9%)	61 (29.1%)	0.382
Alendronate	105 (27.2%)	61 (29.1%)	0.701
Zoledronic acid	40 (10.4%)	13 (6.2%)	0.119
Denosumab	174 (45.1%)	113 (53.8%)	0.051
Teriparatide	58 (15.0%)	51 (24.3%)	0.007
Comorbidity			
Osteoarthritis	267 (69.2%)	142 (67.6%)	0.766
Rheumatoid arthritis	33 (8.6%)	14 (6.7%)	0.512
Diabetes mellitus	92 (23.8%)	81 (38.6%)	<0.001
Hypertension	189 (49.0%)	153 (72.9%)	<0.001
Stroke	61 (15.8%)	55 (26.2%)	0.003
Cataract	106 (27.5%)	64 (30.5%)	0.494
Hyperthyroidism	30 (7.7%)	16 (7.6%)	1.000
Chronic liver disease	70 (18.1%)	52 (24.8%)	0.070

Continuous data were expressed in mean ± SD. Categorical data were expressed in number and percentage. Mann–Whitney U test. Chi-Square test. BMI, body mass index.

**Table 3 ijerph-19-00197-t003:** Cox regression analysis for mortality risks in participants who had discontinued anti-osteoporosis medication.

Variables	Univariate			Multivariate		
	HR	95% CI	*p* Value	HR	95% CI	*p* Value
Gender (Male vs. Female)	2.16	(1.62–2.90)	<0.001	1.87	(1.22–2.86)	0.004
Age (years)	1.06	(1.05–1.08)	<0.001	1.02	(1.00–1.04)	0.078
Number of fractures						
0	reference			reference		
1	1.85	(1.08–3.15)	0.024	1.71	(0.72–4.04)	0.222
≥2	2.97	(1.68–5.23)	<0.001	1.84	(0.74–4.57)	0.188
Bone mineral density (g/cm^2^)						
A: Lumbar spine ≥ 0.715 and right femoral neck ≥ 0.550	reference			reference		
B: Lumbar spine < 0.715 and right femoral neck ≥ 0.550	1.59	(1.07–2.36)	0.021	1.68	(0.93–3.04)	0.083
C: Lumbar spine ≥ 0.715 and right femoral neck < 0.550	2.60	(1.82–3.72)	<0.001	1.90	(1.13–3.14)	0.016
D: Lumbar spine < 0.715 and right femoral neck < 0.550	3.08	(2.13–4.45)	<0.001	3.09	(1.84–5.21)	<0.001
	*p* for trend <0.001			<0.001		
Laboratory test						
Calcium (mg/dL)	0.79	(0.72–0.87)	<0.001	0.87	(0.76–0.99)	0.033
Creatinine (mg/dL)	1.36	(1.22–1.51)	<0.001	1.22	(1.07–1.39)	0.002
Comorbidity						
Diabetes mellitus	1.72	(1.29–2.25)	<0.001	1.44	(0.98–2.11)	0.063
Rheumatoid arthritis	0.74	(0.43–1.27)	0.274	0.84	(0.36–1.97)	0.693
Hyperthyroidism	0.96	(0.57–1.59)	0.861	1.58	(0.78–3.22)	0.207

## Data Availability

The data that support the findings of this study are available from the corresponding author upon reasonable request.

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
