# Peer review of "Low Bone Mineral Density and Calcium Levels as Risks for Mortality in Patients with Self-Discontinuation of Anti-Osteoporosis Medication"

_ijerph, 2021, doi:10.3390/ijerph19010197_

Round 1
Reviewer 1 Report
The manuscript presents an interesting topic. Generally, the manuscript was well designed and it is well written. However, I have some issues:
- What value of Ca concentration in serum was used to criterium of hypocalcemia?
- In the discussion, the Authors should more Focus on the relation between causes of death of patients and low BMD and hypocalcemia.
- In my opinion, the important factor influenced on obtained results may have nutritional factors and dietary intake of calcium. The lack of nutritional factors is a limitation of this study.
Author Response
Point 1. What value of Ca concentration in serum was used to criterium of hypocalcemia?
Response 1: Thank you for the comment. The normal range of serum calcium levels in our study was defined as 8.4-10.2 mg/dL. We have added normal reference ranges of calcium, 25(OH)D3, phosphorus, and intact-PTH in our study (please see lines 124-125). We also have modified the description of “hypocalcemia” to “low calcium levels” in the revised manuscript.
Point 2. In the discussion, the Authors should more Focus on the relation between causes of death of patients and low BMD and hypocalcemia.
Response 2: Thank you for the valuable suggestion. We agree with the reviewer. We have added discussion the relation between causes of death of patients and low BMD (please see lines 271-295) and low serum calcium levels (please see lines 297-307), respectively.
Point 3. In my opinion, the important factor influenced on obtained results may have nutritional factors and dietary intake of calcium. The lack of nutritional factors is a limitation of this study.
Response 3: Thank you for bringing up this question. We agree with the reviewer that nutritional factors are important factors influencing osteoporosis, but this information was not available in our database. We have also stated this as a limitation (please see lines 317-318).
Reviewer 2 Report
My assessment is attached

Reviewer 3 Report
General:
- Why are you only focussing on self-discontinuations of anti-OP medications? Patients with drug holidays might also loss BMD and are definitely worth to be analyzed. Please explain this properly and/or include drug holiday patients.
Abstract: BMD = Bone Mineral Density
Methods
Study design:
- You are writing "...absence of Deno claim more than 6m+8w…" What about potential switches from Deno to any other OP medication, this patients would be included as well? Since an absence of Deno does not necessarily mean a discontinuation of OP medication. Please specify what happend to switchers, because for Teri you are writing "...absence of any anti-op medication…"
- What about bisphosphonate / SERM users, what was the definition for discontinuation? Please include.
Fracture Identification:
- How did you ensure that all these fractures were low-trauma fractures?
Statistical Analysis:
- Please describe which variables were included into the multivariate model
Results
Demographic data and concomitant medication by BMD
- Please provide mean / median follow-up of included patients
- Please provide information on participants per age groups, since your inclusion criterium was aged 20+ years and I am wondering how many "young" patients were in the analyzed groups
Predictive factors for mortality in patients with prior exposure to anti-osteoporosis medication that was later discontinued - Table 3
- Why have you only analyzed Diabetes, what about the other comorbidites? I guess the comorbidities have a high impact on the mortality. Please explain this and/or included them into the model.
- Which variables were included into the multivariate model? Have you adjusted on the comorbidities?
- Why have you excluded the discontinued drugs from this analyze? This makes no sense for me, especially since you are interested in the BMD and we know that discontinuation of Deno might lead to a rebound effect with multipe vert fx, whereas BPs are pretty safe to be discontiued? Please explain this and/or included them into the model.
Figure 1C
- What happend to male group D?
Conclusions
- As you said, you can not see causal relationships, so please change the conclusion that it sounds more soft and less final.
- For instance:
Hypocalcemia was also associated with a higher mortality risk
Author Response
General:
- Why are you only focussing on self-discontinuations of anti-OP medications? Patients with drug holidays might also loss BMD and are definitely worth to be analyzed. Please explain this properly and/or include drug holiday patients.
Author’s response: Thank you for the valuable comments. Patients may discontinue osteoporosis medication due to drug holiday or self-discontinuation. A prior report found that mortality risks among patients continue to take bisphosphonate are similar to those of patients on bisphosphonate drug holidays. Therefore, our study aimed to investigate mortality risks in patients who self-discontinued anti-osteoporosis medication. We have revised the introduction accordingly (please see line 67-72, 85-87 and reference 11).
Methods
Study design:
- You are writing "...absence of Deno claim more than 6m+8w…" What about potential switches from Deno to any other OP medication, this patients would be included as well? Since an absence of Deno does not necessarily mean a discontinuation of OP medication. Please specify what happend to switchers, because for Teri you are writing "...absence of any anti-op medication…"
Author’s response: Thank you for bringing up this important question. We would like to take this opportunity to clarify our exclusion criteria for denosumab users. We excluded patients switching from denosumab to any anti-osteoporosis medication other than denosumab within 6 months + 8 weeks after prior denosumab treatment (please see lines 110-113).
- What about bisphosphonate / SERM users, what was the definition for discontinuation? Please include.
Author’s response: Thank you for this query. We have added the definition of bisphosphonate or raloxifene discontinuation as ≥12 months without any anti-osteoporosis medication prescription claims after prior bisphosphonate or raloxifene treatment (please see lines 114-116).
Fracture Identification:
- How did you ensure that all these fractures were low-trauma fractures?
Author’s response: Thank you for bringing up this important question. We excluded fractures due to vehicular accidents or high-impact trauma (ICD-9 codes E810-E819, E881-E883, and E8841), pathological fractures (ICD-9-CM codes 733.14 and 733.15) and patients with a diagnosis of Paget’s disease (ICD-9-CM code 731.0) (please see line 137-140).
Statistical Analysis:
- Please describe which variables were included into the multivariate model
Author’s response: Thank you for the comments. The adjustment of age, gender, number of fractures, BMD, laboratory tests and comorbidities were included in the cox regression model (please see lines 165-166).
Results
Demographic data and concomitant medication by BMD
- Please provide mean / median follow-up of included patients
Author’s response: Thank you for the valuable input. The mean and median follow-up time of patients included were 5.0± 0.2 years and 4.4 years respectively (please see line 171).
- Please provide information on participants per age groups, since your inclusion criterium was aged 20+ years and I am wondering how many "young" patients were in the analyzed groups
Author’s response: Thank you for the important question. To avoid inclusion of participants with osteoporosis due to diverse etiologies, we have revised the study design to exclude patients younger than 45-year-old were excluded for decreasing the bias related to heritable bone disorders (please see line 98 and revised Table 1, 2 & 3).
Predictive factors for mortality in patients with prior exposure to anti-osteoporosis medication that was later discontinued - Table 3
- Why have you only analyzed Diabetes, what about the other comorbidites? I guess the comorbidities have a high impact on the mortality. Please explain this and/or included them into the model.
Author’s response: Thanks for the suggestion. As reviewer’s suggestions, we have included comorbidities of rheumatoid arthritis and hyperthyroidism in Cox regression analysis (please see revised Table 3) and revised our manuscript. We found that comorbidities did not seem to be associated with mortality risks among patients with poor adherence.
- Which variables were included into the multivariate model? Have you adjusted on the comorbidities?
Author’s response: We included age, gender, number of fractures, BMD, laboratory tests, and comorbidities (diabetes mellitus, rheumatoid arthritis and hyperthyroidism) in the multivariate model. We revised the method section (please see lines 164-165 and revised Table 3).
- Why have you excluded the discontinued drugs from this analyze? This makes no sense for me, especially since you are interested in the BMD and we know that discontinuation of Deno might lead to a rebound effect with multipe vert fx, whereas BPs are pretty safe to be discontiued? Please explain this and/or included them into the model.
Author’s response: Thank you for the question. We agree with the reviewer’s opinion that types of discontinued medication might also contribute to the mortality risks. However, in Table 2, we did not observe a significant difference of discontinued drugs between survival and mortality groups. Therefore, discontinued drugs were not included in the Cox regression analysis.
Figure 1C
- What happend to male group D?
Author’s response: Thank you for the comment. The sample size of male group D was relatively small (only 15 people) and the follow-up period was less than 4 years. This explained a significant drop of male group D curve. We added the small sample size of male participants as a limitation (please see line 311).
Conclusions
- As you said, you can not see causal relationships, so please change the conclusion that it sounds more soft and less final.
- For instance:
Hypocalcemia was also associated with a higher mortality risk
Author’s response: Thank you for the suggestion. We have revised the title of this manuscript and conclusion statement accordingly (please see lines 323-325).
Round 2
Reviewer 2 Report
The changers are convincing
Reviewer 3 Report
Dear authors,
thank you for this interesting manuscript and the revision. I am fine with alle changes and your comments.